

# Forecasting temperature and rainfall using deep learning for the challenging climates of Northern India

Syed Nisar Hussain Bukhari[1] and Kingsley A. Ogudo[2]

[1] National Institute of Electronics and Information Technology (NIELIT), Srinagar, J&K, India
[2] Department of Electrical & Electronics Engineering Technology, University of Johannesburg, Johannesburg, South Africa

Corresponding author
Syed Nisar Hussain Bukhari,
nisar.bukhari@gmail.com

## ABSTRACT

Accurate temperature and rainfall (T&R) forecasting is vital for the climate-sensitive regions of Northern India, particularly Jammu, Kashmir, and Ladakh, where volatile weather patterns significantly affect livelihoods, socio-economic development, and disaster management efforts. Despite their importance, traditional forecasting methods often fall short due to their high computational demands and inability to provide localized, real-time predictions, leaving a critical research gap in addressing these challenges. This study addresses the need for precise and efficient T&R forecasting using deep learning-based framework tailored to the unique climatic conditions of these regions. The major research focus is to develop and evaluate a model capable of capturing complex temporal dependencies in localized time-series weather data. Utilizing data from the Indian Meteorological Department (IMD) for Jammu, Srinagar, and Ladakh stations covering the period from January 1, 2000, to December 31, 2023, the proposed framework employs recurrent neural networks (RNN) and long short-term memory (LSTM) architectures, both optimized for time-series forecasting. Key findings reveal that while both RNN and LSTM models exhibit robust performance in single input single output (SISO) setups, RNN model consistently outperforms the LSTM in capturing intricate temporal relationships. The RNN model in MIMO configuration achieved significantly lower mean absolute error (MAE), root mean squared error (RMSE), and mean squared error (MSE) for Jammu, Srinagar, and Ladakh, with respective values of [0.0636, 0.1011, 0.0401] for Jammu, [0.1048, 0.1555, 0.0455] for Srinagar, and [0.0854, 0.1344, 0.0411] for Ladakh. These results underscore the RNN model's precision, making it a practical tool for real-time weather forecasting. By enhancing the accuracy of T&R predictions in regions with challenging meteorological conditions, this study contributes to improved climate adaptation strategies, disaster preparedness, and sustainable development. Its findings hold broader implications for advancing localized forecasting technologies in other regions with similar climatic complexities.

## INTRODUCTION

Jammu & Kashmir (J&K) and Ladakh, located in the western Himalayas, are India's northernmost union territories characterized by highly variable and extreme weather.

These unpredictable climatic conditions significantly affect agriculture, infrastructure, and socio-economic development in the region. Spanning a combined area of 222,236 square kilometers, these territories lie between latitudes 32°17′N to 37°05′N and longitudes 72°31′E to 80°20′E (*Balasubramanian, 2016*; *P. Office of Additional Director General of Meteorology (Research), 2014*). Topographically, the region is dominated by mountains, with four major Himalayan ranges: Karakoram, Zanskar, Ladakh, and Pir Panjal. The Karakoram range in the north includes K2 (8,611 m), the world's second-highest peak, and other notable summits like Gasherbrum (8,570 m) and Masharbrum (7,827 m) (*Kashmir Travels, 2024*). The Zanskar range, part of the Tethys Himalaya, separates the Indus and Kashmir valleys, with elevations reaching up to 6,000 m. The Ladakh range, north of the Indus River, merges with the Kailash range in Tibet and includes peaks above 5,000 m (*P. Office of Additional Director General of Meteorology (Research), 2014*). To the south, the Pir Panjal range (approx. 4,100 m) forms a climatic barrier between the Jammu and Kashmir regions. Jammu features plains and forested hills merging into the Shivalik and Pir Panjal ranges (*Ali et al., 2023*). The Kashmir valley, lying between Pir Panjal and Zanskar ranges, is surrounded by snow-capped mountains rising up to 4,900 m, and has an average elevation of 1,615 m (*P. Office of Additional Director General of Meteorology (Research), 2014*). Ladakh, a cold desert with elevations averaging above 3,650 m, accounts for nearly two-thirds of the region's total area (*P. Office of Additional Director General of Meteorology (Research), 2014*). This varied topography results in sharp climatic gradients over short distances, complicating weather prediction. Several perennial rivers, including the Indus, Jhelum, and Chenab, originate in these mountains and sustain the region's hydrology.

The meteorological profile of the Jammu, Kashmir, and Ladakh regions is marked by significant spatial and temporal variability due to their diverse topography and altitude gradients. Jammu experiences a subtropical climate with hot summers, mild winters, and substantial monsoon rainfall between June and September (*Kashmir Travels, 2024*). In contrast, the Kashmir Valley features a temperate climate, characterized by cold winters, moderate summers, and two major precipitation periods—winter snowfall from western disturbances (December to March) and moderate rainfall during spring and autumn. Ladakh, situated at a much higher altitude, falls under a cold desert climate with extremely low annual precipitation (less than 100 mm on average) and large diurnal temperature variations (*Ali et al., 2023*). The region receives minimal monsoonal influence and is instead affected by western disturbances, particularly in the form of snowfall. Seasonal and interannual variability in both temperature and precipitation is pronounced across these regions, presenting significant challenges for reliable forecasting. This meteorological diversity necessitates models capable of capturing complex, localized patterns in both short- and medium-term forecasts.

Accurate weather forecast plays a crucial role in various sectors, including agriculture, tourism, and construction. In addition, will aid famers in crop planning, risk mitigation, and water resource management for optimized yields and resilience. Numerical weather prediction (NWP) is a process of simulating weather conditions, integrating observed data and running simulations to project upcoming weather patterns. This involves solving

complex systems of non-linear mathematical equations, derived from specific mathematical models, with the help of powerful supercomputers (*Frnda et al., 2022*). Meteorological characteristics like temperature and precipitation gathered from a specific location over a period of time yield quantitative data. This quantitative data can be utilized by NWP systems to gain insights into the scientific principles governing atmospheric processes and to make predictions about future atmospheric conditions. However, the performance of NWP is restricted by several factors like extensive resource utilization, inherent uncertainties and limited model resolution. On the other hand, reliability of statistical methods like auto-regressive moving average (ARMA) and its variants depends on the data quality and their ability to effectively depict time-series (TS) data with non-stationary behaviors across seasons (*Poornima & Pushpalatha, 2019*).

Over the past few decades, deep learning (DL) techniques have proven as a transformative force in advancing the accuracy and efficiency in weather and rainfall forecasting (*Waqas et al., 2023*). According to *Hayati & Mohebi (2007)*, *Waqas et al. (2024)* data-driven modeling systems can be employed to decrease the computational requirements of NWP processes. This can be ascribed to their capacity of estimating the intrinsic patterns and dynamics of TS data even without prior knowledge of the parameters, while also accounting for the uncertainty present in observations and system noise. DL techniques such as recurrent neural networks (RNNs) and long short-term memory (LSTM) models are regarded as better suited for handling weather data due to their strong capability to effectively process TS data (*Kratzert et al., 2018*; *Kashiwao et al., 2017*). By incorporating input, output, and forget gates, LSTM-RNN achieves a network that can effectively maintain state and propagate gradients over extended time periods in a stable manner.

This study proposes a DL based predictive modeling framework for forecasting T&R that will benefit the local community and stakeholders of J&K and Ladakh by helping them adapt to the unique weather conditions of the area, making their daily life more predictable and safe. The model will also aid the governing bodies of J&K and Ladakh by enhancing foreplanning, risk mitigation, and resource management, ultimately contributing to the economic growth and development of the local population. The proposed model utilizes past values of local temperature and rainfall (T&R) of J&K and Ladakh within a defined time-step as input to forecast the same for the upcoming days. Using weather data of past 23 years from local weather stations will allow the model to capture and model the unique properties exhibited by the weather of J&K and Ladakh.

## Contributions

This work introduces a novel T&R forecasting model that demonstrates robust adaptability to evolving and extreme weather trends in the climatically complex regions of Jammu & Kashmir and Ladakh. The proposed approach leverages DL architectures, specifically Recurrent Neural Networks (RNNs) and Long Short-Term Memory (LSTM) networks, to effectively model temporal dependencies in weather data. Both single-input single-output (SISO) and multi-input multi-output (MIMO) frameworks are explored to assess model performance under varying forecasting scenarios. By reducing reliance on

traditional Numerical Weather Prediction (NWP) methods, the study significantly lowers computational costs and addresses the limitations of conventional approaches that often fail to capture localized climatic variations. In addition, the proposed models are rigorously benchmarked against existing works, providing valuable insights into their comparative effectiveness and potential for enhancing region-specific weather forecasting accuracy.

## Motivations

RNNs and their advanced variant LSTM networks, have shown promise in time series forecasting tasks, including weather prediction (*Waqas & Humphries, 2024*). However, most existing studies applying these models focus on regions with relatively stable climatic behavior and less topographical diversity. The union territories of Jammu, Kashmir, and Ladakh, by contrast, exhibit highly complex terrain and a wide range of weather variability, posing unique challenges for temporal modeling. In this context, it becomes imperative to evaluate the suitability and robustness of these sequential DL architectures. This study is motivated by the need to understand how RNN and LSTM models respond to such climatic complexity, particularly in capturing long-term dependencies and handling nonlinear fluctuations inherent in the region's meteorological data. A comparative analysis of these models under both SISO and MIMO frameworks is undertaken to provide a nuanced understanding of their strengths and limitations in geographically and climatically diverse settings.

## RELATED WORK

Over time, various techniques have been developed to forecast weather conditions, particularly using T&R. A study by *Han et al. (2021)* trained and tested several neural networks, including feedforward neural networks (FFNN), recurrent neural networks (RNN), long short-term memory (LSTM), and gated recurrent units (GRU), to create a model capable of generating weather predictions reflective of local conditions. Using airport data from the National Solar Radiation Database (NSRDB) and validating it with on-site measurements, GRU model achieved the lowest mean squared error (MSE) of 2.96 among all models. In a separate study aiming to improve weather forecasts for five cities in China, *Nketiah et al. (2023)* proposed five multivariate time series models specifically designed for atmospheric temperature prediction, based on RNN architectures. The experimental results showed that using the LSTM-RNN model yielded the lowest prediction error for atmospheric temperature compared to baseline models. A new DL model called the spatial feature attention long short-term memory (SFA-LSTM), introduced by *Suleman & Shridevi (2022)*, effectively captured both spatial and temporal relationships among multiple meteorological features for temperature forecasting. Comparative analysis with baseline models demonstrated that the SFA-LSTM achieves state-of-the-art prediction accuracy, offering the additional benefit of enhanced interpretability of spatial features. To capture localized patterns and dependencies contributing to fluctuations in maximum and minimum temperatures, a study by *Karevan & Suykens (2020)* introduced the transductive-LSTM (T-LSTM) model. This approach, by taking surrounding data points into account, allowed for more accurate temperature

forecasts by leveraging the context provided by nearby data points. In another study, *Mi (2023)* used summer temperature data from Seoul, Korea, spanning from 2013 to 2017, incorporating input parameters such as average temperature, solar radiation, average relative humidity, average wind speed, and average latent heat flux. This study compared the predictive capabilities of linear regression and LSTM models for daily average temperature forecasting, finding that the LSTM model had a lower mean absolute error of 0.19 compared to the linear regression model's 0.89. To make use of the rapid data processing capabilities of CNNs with the memory function of LSTMs for time series temperature data, *Guo (2023)* proposed a hybrid CNN-LSTM model. Experimental results showed that this hybrid model outperformed individual CNN and LSTM models, achieving higher temperature prediction accuracy. A similar study by *Mung & Phyu (2023)* used three DL models—CNN, LSTM, and a CNN-LSTM ensemble—to predict features such as minimum temperature, maximum temperature, humidity, and wind speed. Using root mean square error (RMSE) as the comparison metric, the study found that the ensemble model achieved the lowest RMSE compared to the individual models. Research by *Nugraha, Ariawan & Arifin (2023)* explored LSTM's application in weather forecasting using a dataset from the Serang Maritime Meteorological Station, covering variables such as temperature, humidity, sunshine, and wind speed from January 1, 2018, to October 28, 2023. Their LSTM model achieved an RMSE of 0.37 for temperature, 0.72 for wind speed, 2.79 for sunlight, and 5.05 for humidity In another study, *Fan et al. (2022)* developed a generalized CNN-LSTM model using an encoder-decoder architecture to predict temperature, humidity, rainfall, and wind speed. Trained on the IEEE Big Data IARAI's Weather4cast 2021 dataset, the model showed a significant reduction in loss as assessed by MSE values. The study by *Chen, Huang & Yang (2023)* proposed an LSTM model integrating multiple linear regression and Pearson's correlation coefficients to improve weather prediction for aviation safety. This model, trained on a numerical dataset of 10 weather parameters with key features (sea pressure, dew point temperature, and relative humidity) identified through feature selection, improved its forecasting accuracy from an RMSE of 4.0274 to 2.2215 and a MAPE reduction from 23.0538% to 5.0069%. A comparison study by *Gong et al. (2022)* evaluated ConvLSTM (LSTM with convolutional filters) against an advanced Stochastic Adversarial Video Prediction (SAVP) network for temperature forecasting. The models were assessed using metrics like MSE, anomaly correlation coefficient (ACC), and structural similarity index (SSIM), revealing that the SAVP model outperformed ConvLSTM with an MSE of 2.3, ACC over 0.85, and SSIM around 0.72. Lastly, study *Li et al. (2023)* employed LSTM and its deep variant, Deep LSTM (DLSTM), to forecast daily air temperature. Trained on 1,097 weather data points from central and southern regions of Tabriz, Iran, from 2017 to 2019, the DLSTM model achieved an RMSE of 0.08 and an R-squared value of 0.99.

In summary, although prior studies have demonstrated the utility of ML and DL techniques in weather forecasting, they often lack contextual adaptability, overlook complex terrains, or fail to comprehensively evaluate different modelling configurations. This study addresses these limitations by focusing on the climatically diverse Himalayan regions of Jammu, Kashmir, and Ladakh, and by comparing both SISO and MIMO

LSTM-based architectures using long-term T&R datasets. In doing so, it offers deeper insights into model suitability and forecasting performance under highly variable geographic and climatic conditions.

## MATERIALS AND METHODS

The model flowchart used in this study is presented in Fig. 1. It includes T & R data collection and preprocessing phase, model configuration (*Hochreiter & Schmidhuber, 1997*; *Werbos, 1988*), training and evaluation. The final stage of the experimentation involves assessing and comparing the performance of the trained models.

### Data collection

This study utilize meteorological data sourced from the Indian Meteorological Department (IMD) website (https://mausam.imd.gov.in/), specifically for Jammu & Srinagar and Ladakh weather stations. The dataset for each station comprises daily time-series data with 8,401 data points, covering the period from January 1, 2000, to December 31, 2023. It includes various weather variables such as rainfall, minimum temperature, and maximum temperature. Rainfall is measured in millimeters (mm), while temperatures are recorded in Celsius. These meteorological features were chosen for their ability to provide valuable insights into current weather conditions at specific locations and times, aiming to capture the weather state accurately. The data was obtained by submitting a request *via* the IMD Pune Data Supply Service portal "https://dsp.imdpune.gov.in/data_supply_service.php" specifying the required parameters, such as location, time period, and frequency, following the standard procedure outlined by IMD.

### Data preprocessing

To ensure the quality and suitability of the time-series data for deep learning models, several preprocessing steps were applied. These steps aimed to reduce noise, handle trends, normalize scale differences, and prepare structured input-output sequences for model training and evaluation. Each step is described below along with its rationale.

#### Smoothing

**Rationale:** Real-world time-series data often contain short-term fluctuations that can obscure underlying patterns. Smoothing helps in highlighting long-term trends and reducing noise.

**Method:** An exponential moving average (EMA) technique was applied to each time-series. EMA (*Welles, 1978*) is the weighted average value where weights are decreased gradually such that more importance is given to recent data points as compared to the historical ones, or vice versa. EMA changes at a faster rate and is more sensitive to the data points. Mathematically, EMA of a data point is calculated using Eq. (1).

$$EMA_t = \begin{cases} x_0 & t = 0 \\ \alpha x_t + (1 - \alpha)EMA_{t-1} & t > 0 \end{cases} \quad (1)$$

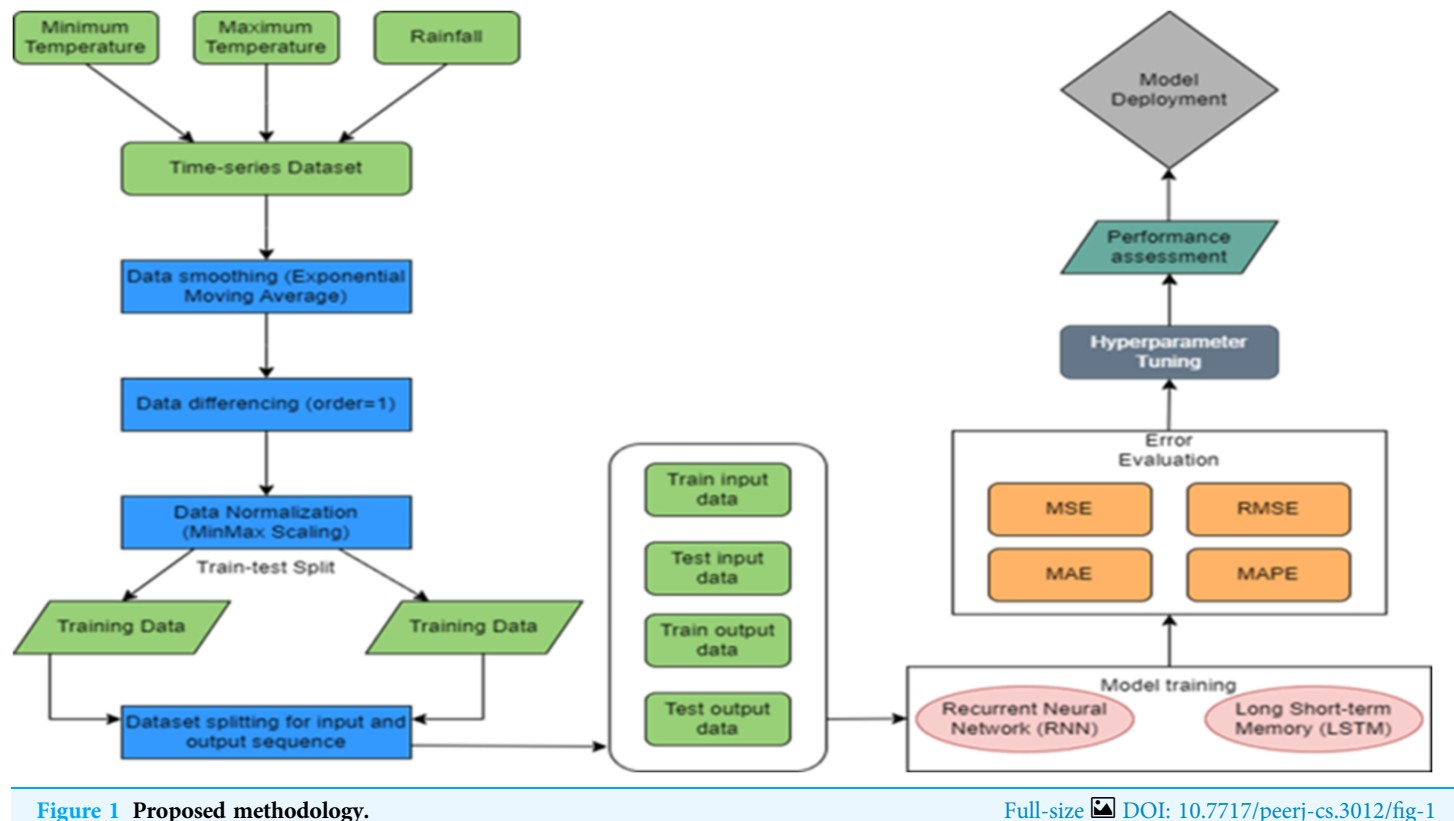

**Figure 1 Proposed methodology.**

where $\alpha$ is a smoothing factor and its value lies between 0 and 1. It represents the weight applied to the very recent period. For this study, the value of $\alpha$ has been set as 0.5.

### Differencing

**Rationale:** Most time-series forecasting models assume stationarity. Differencing removes trends and stabilizes the mean, making the data more suitable for modeling.

**Method:** The first-order differencing technique was employed to ensure stationarity of the weather time which involves taking the difference between each data point and its immediate predecessor, effectively removing underlying trends. By focusing on short-term changes rather than absolute values, differencing enhances the stability of statistical properties over time and is mathematically expressed in Eq. (2).

$$\Delta y_t = y_t - y_{t-1}. \tag{2}$$

Here, $\Delta y_t$ represents the differenced value at time t, $y_t$ is the original data point at time t, and $y_{t-1}$ is the value at the previous time step.

### Normalization

**Rationale:** The original variables have different units and ranges, which can negatively affect model convergence and performance. Normalization ensures consistent scale and helps activation functions in deep learning models to work more effectively.

**Method:** Min–Max scaling was applied to rescale each feature to a [0,1] range between 0 and 1 (*Tukey, 1962*) Mathematically Min-Max scaling is represented by Eq. (3).

$$X_t = \frac{X_t - X_{min}}{X_{max} - X_{min}} \tag{3}$$

where $X_t$ represents the data point at time 't' in time-series, $X_{max}$ and $X_{min}$ represent maximum and minimum data points in the sequence respectively.

### Train-test split

**Rationale:** Separating the dataset into training and testing portions is essential to evaluate model generalization.

**Method:** The dataset was split into 80% training and 20% testing subsets. While the training set allows the model to capture and learn various relationships and patterns in data, the testing set is used to assess the model's generalizing ability. Each weather station training dataset consists of 6,720 rows containing weather data from 2000-01-01 to 2018-05-25, while the testing dataset contains 1,680 rows spanning from 2018-05-26 to 2023-12-31.

### Input-output sequence generation

**Rationale:** Time-series data require conversion into supervised learning format, where the past values (inputs) are mapped to future predictions (outputs).

**Method:** A moving window algorithm with a window size of 7 was used to generate input-output pairs. The window size was set to 7 to reflect a 1-week temporal context, which is commonly used in meteorological time series to capture short-term climatic trends. Each sequence included three temperature variables as features. The input sequence consisted of past 7-day values, while the output predicted the target variable for the forecast horizon.

## Models used in the current study

The proposed T&R model leverages advanced DL techniques, specifically RNN and LSTM, to improve prediction accuracy. The following subsections offer a detailed explanation of the RNN and LSTM models.

### Recurrent neural network

A traditional recurrent neural network (RNN) (*Hochreiter & Schmidhuber, 1997*) can be viewed as an enhanced version of a feed-forward NN, designed with an internal memory mechanism to process sequential inputs. RNNs are capable of capturing dependencies from earlier elements in a sequence, allowing them to predict cumulative metrics from sequences of varying lengths, particularly those involving a time-based component (*Bukhari & Pandit, 2024*). They do this by preserving information in a sequential order *i.e.*, at each time step, the RNN receives the current input along with contextual information passed forward through its feedback connections.

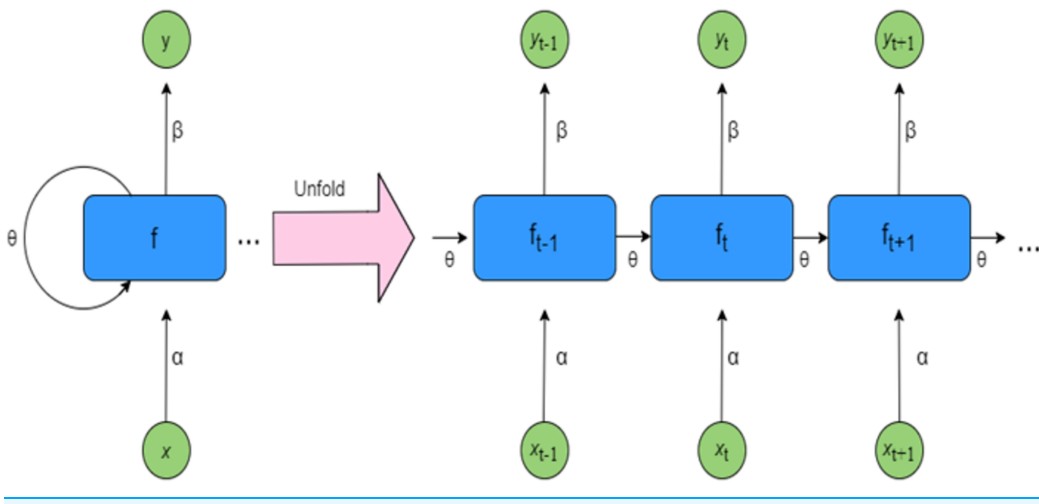

**Figure 2** **Architecture of RNN.**

Figure 2 presents a simple RNN architecture, where inputs from time step t to t+1 are denoted as $x_t$ to $x_{t+1}$, and the corresponding outputs are represented as $y_t$ to $y_{t+1}$. The hidden states, labeled $f_{t-1}$ to $f_{t+1}$, capture the intermediate information, while the weights for the hidden states, inputs, and outputs are represented by $\theta$, $\alpha$, and $\beta$ respectively. RNNs utilize shared parameters across time steps, meaning the same set of weights is applied throughout each layer of the network. Despite being shared, these weights are still refined during training through backpropagation and gradient descent, which helps the model improve its learning performance over time (*Bukhari, 2024*). A detailed explanation of RNN functionality is provided in Algorithm S1. In RNNs, activation functions are applied solely to the hidden layers. The input layer simply receives and passes the data forward without performing any computation. Without non-linear activation functions, even deep NNs would behave like extended linear regression models, lacking the capability to learn complex patterns from real-world data. To overcome this, commonly used activation functions include ReLU (Rectified Linear Unit), tanh, Sigmoid, and Softmax.

- **Backpropagation through time**

Backpropagation through time (BPTT) is an advanced form of the traditional backpropagation algorithm, specifically adapted for training RNNs. In BPTT, the RNN is unfolded across its time steps, allowing gradients to be propagated backward through this temporal sequence (*Bukhari & Pandit, 2024*). This approach enables the model to capture and learn sequential dependencies by adjusting its weights based on the cumulative error over multiple time steps (*Hochreiter & Schmidhuber, 1997*). A simplified overview of the BPTT process is outlined in Algorithm 2 (provided as Algorithm S2). A simplified breakdown of the steps involved in BPTT is given in Algorithm 2 (provided as Algorithm S2).

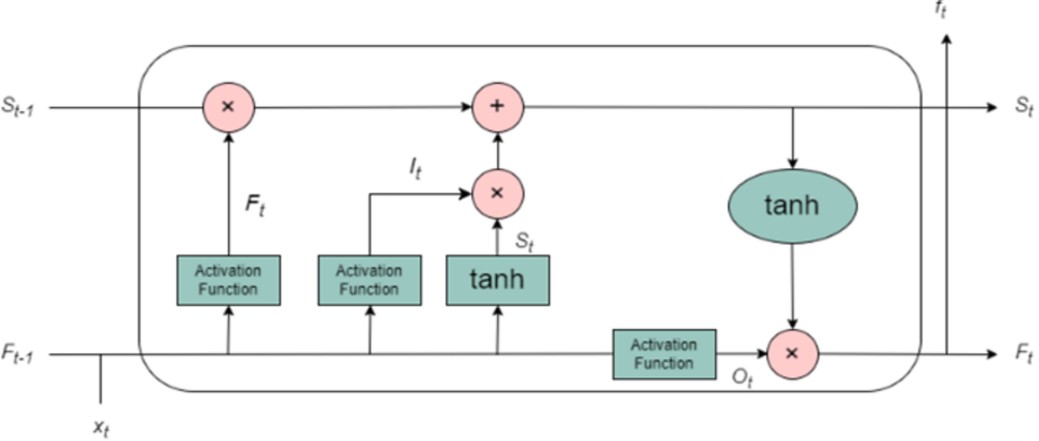

**Figure 3 Architecture of LSTM.**

### Long short-term memory

The limitations of traditional recurrent neural networks (RNNs) such as the vanishing gradient problem, which makes it difficult for the network to capture long-term dependencies, and the exploding gradient problem, which can cause instability during training are mitigated by the use of LSTM networks. LSTM, an advanced variant of RNN, was designed specifically to overcome these challenges, allowing the network to effectively learn from long-term sequential data. LSTM units are structured around a specialized memory cell, which serves as the core of the architecture. This memory cell allows the network to maintain and update information over long time periods, enabling it to better understand the temporal dependencies in the data. The memory cell is updated and modified by three primary gating mechanisms: the input gate, the forget gate, and the output gate as shown in Fig. 3. These gates function collectively to control the flow of information into, out of, and within the memory cell, making LSTM particularly effective at handling time-series data such as T&R sequences. The input gate controls how much of the incoming data at each time step (*i.e.*, the current input and the previous hidden state) should be incorporated into the memory cell. It is a sigmoid function that outputs values between 0 and 1, determining which values will be updated in the cell state. The stronger the activation of the input gate, the more new information will be incorporated. The forget gate, another sigmoid function, determines what portion of the previous cell state should be discarded or retained. Essentially, it tells the network which past information is still relevant to be remembered and which can be forgotten. This is particularly important in time-series forecasting, where certain past events may have a more significant influence on future outcomes than others. By selectively forgetting less useful information, the LSTM can focus on more relevant data points. The output gate governs the flow of information from the memory cell to the output of the LSTM unit. It computes the hidden state, which is a function of the current memory cell and the output from the input gate. The hidden state is then passed to subsequent time steps and contributes to the final prediction. It is

also used in the subsequent layers of the network for further processing. This gating mechanism enables the LSTM to learn what information to remember and what to forget across different time scales, making it especially suitable for modelling meteorological time series data, where both short-term variability and long-term seasonal trends are critical. The working of LSTM is explained using algorithm 3 (provided as Algorithm S3).

## Model configuration and development

The proposed models have been trained using two distinct configurations: SISO and MIMO (*Oduro-Gyimah & Boateng, 2019*). In a SISO setup, the model uses a single input variable to predict its future values, requiring separate models for each parameter. MIMO, in contrast, processes multiple inputs simultaneously and forecasts all outputs at once, making it more efficient with only one model per weather station is needed. Based on this, three RNN and three LSTM models were built using MIMO for three stations, while nine RNN and nine LSTM models were trained using SISO (three per station for different temperature variables). Each model was trained with carefully tuned hyperparameters like layers, hidden units, and learning rate which, unlike model parameters, remain fixed during training but greatly influence performance (*Bukhari, 2024*). The hyperparameters tuned in this study include number of epochs (epochs), batch size (batch_size), optimization algorithms (optimizer) and activation functions (activation). The technique chosen for tuning hyperparameters is GridSearchCV that automates the process of trying various hyperparameter configurations. For both RNN and LSTM models epochs represent the number of times the model sees the full dataset during training each involving a forward and backward pass. Fewer epochs can lead to underfitting, while too many may cause overfitting. To ensure a fair comparison, both models were trained using 50 to 300 epochs, a range selected through preliminary testing and GridSearchCV tuning (*Bukhari & Pandit, 2024*). While LSTM generally benefits from longer training, early stopping was applied when needed to prevent overfitting. The batch size, which defines how many samples are processed per update, was also tuned to optimize performance.

To achieve an optimal balance between performance and efficiency, experiments were conducted using batch sizes of 16, 32, 64, and 128. Different activation functions, including ReLU, tanh, and sigmoid, were evaluated to determine the most effective nonlinear transformation for each model. Both RNN and LSTM architectures were configured with a single hidden layer, and the number of units in this layer ranged from 32 to 128 (*Bukhari, 2024*). The Adam optimizer was primarily chosen due to its strong convergence properties, with the learning rate tuned within the range of 0.001 to 0.01. For comparative purposes, additional training runs were performed using the SGD optimizer (with momentum set to 0.9) and the RMSprop optimizer (*Bukhari & Pandit, 2024*). To prevent overfitting and promote better generalization, dropout regularization was applied, with dropout rates varying from 0.1 to 0.3 depending on the model's complexity and configuration. Final model configurations were selected based on their best performance on the validation set and were subsequently assessed using standard error metrics such as MSE, MAE, RMSE, and MAPE as described in the next section.

*Computing environment*

The experiments were conducted on a computing system equipped with an Intel Xeon Silver 4,214 processor, 128 GB RAM, and an NVIDIA RTX 3090 (24 GB VRAM) GPU. The operating system used was Windows 11 Pro (64-bit), and all DL models were implemented using Python 3.8 with TensorFlow 2.9.

## Performance metrics

The models underwent repeated training on the training data and were then evaluated using the testing data. The RNN and LSTM models, in both SISO and MIMO configurations, were tested on a test dataset consisting of 1,680 input sequences. These models process test data in a sequential manner, one time step at a time, update their internal states and generate predictions for future time step. The generated output that represents the future value of the temperature variables has been compared against the actual output values present in test dataset using various statistical measures. For evaluating TS regression models the commonly used metrics are mean absolute error (MAE), mean squared error (MSE), root mean squared error (RMSE) and mean absolute percentage error (MAPE) (*Kumar, Kumar & Kumar, 2020*) which are calculated using Eqs. (4), (5), (6) and (7), respectively. MAE is simply the average of differences between predicted value and actual value of the observations in test set. It helps in evaluating the overall performance of the model. MSE refers to the average of squared differences between predicted and actual value of observations while as RMSE can be determined by taking the root of MSE. Due to square function, both MSE and RMSE emphasize larger errors, providing insights into model's reliability. Additionally these metrics are in the same unit as the data and hence can be easily interpreted in the context of the problem domain. MAPE is calculated by dividing the MAE by actual value of the observation and helps in comparing different forecasting models based on their relative performance.

$$MAE = \frac{1}{N} \sum_{i=1}^{N} |y_t - \acute{y}_t| \tag{4}$$

$$MSE = \frac{1}{N} \sum_{i=1}^{N} (y_t - \acute{y}_t)^2 \tag{5}$$

$$RMSE = \sqrt{\frac{\sum_{i=1}^{N} (y_t - \acute{y}_t)^2}{n}} \tag{6}$$

$$MAPE = \sum_{i=1}^{N} \frac{|y_t - \acute{y}_i|}{y_i} \tag{7}$$

where $y_t$ and $\acute{y}_t$ are the actual and predicted values of temperature variables at time step 't'.

# RESULTS AND DISCUSSION

## Result analysis

As mentioned previously, the RNN and LSTM models have been trained using different sets of hyperparameters and tested using various evaluation metrics. The performance of

**Table 1 Comparison of RNN and LSTM in MIMO configuration for Jammu station.**

| Metric | LSTM | RNN |
|---|---|---|
| MAE | 0.1856 | 0.0636 |
| MSE | 0.0805 | 0.0401 |
| RMSE | 0.2594 | 0.1011 |
| MAPE | 0.6710 | 0.3990 |

models in MIMO and SISO configurations has been recorded individually for each weather station using a set of evaluation metrics mentioned in the above sub-section. Table 1 presents a comparison between MIMO LSTM and RNN models based on various metrics for predicting T&R of Jammu station. Tables 2 and 3 provide similar information of other two weather stations- Srinagar and Ladakh, respectively. For each station it has been observed that RNN model outperforms LSTM model in multivariate prediction using MIMO configuration. From Tables 1, 2 and 3 it can be clearly seen that the LSTM consistently exhibited losses approximately double as compared to those exhibited by RNN. The results indicate that RNN in MIMO configuration performs significantly well in capturing the temporal relationships of weather data in contrast to LSTM in same configuration. Analogous observations were recorded when the models were tested in SISO configuration. Table 4 provides a comparison between RNN and LSTM models assessed separately for the three weather parameters of Jammu station. Tables 5 and 6 offer a similar comparative analysis for weather stations of Srinagar and Ladakh respectively. From Tables 5 and 6 it can be observed that for each temperature variable RNN demonstrated superior performance and reduced loss values as opposed to LSTM in the same configuration. All reported errors for the T&R variables are presented in their respective units: MAE and RMSE in degrees Celsius ($°C$) for temperature, and millimeters (mm) for rainfall; MSE in square degrees Celsius ($°C^2$) for temperature, and square millimeters ($mm^2$) for rainfall; and MAPE as a percentage (%), enabling clear interpretation of the forecasting accuracy. Furthermore, it has been noticed that both models perform noticeably well in the SISO configuration across all weather stations. This indicate that the variables present in weather data are relatively univariate and do not possess strong interdependencies between them. Such relationships can be adequately represented using a single input variable to forecast a single output variable.

## Performance analysis with existing techniques

In addition to intra-model comparisons, the best-performing architecture *i.e.*, RNN in the SISO configuration was evaluated against benchmark models reported in the literature, as shown in Table 7. The LSTM-based model proposed in *Mi (2023)* achieved a minimum MAE of 0.19 for daily temperature forecasting. In contrast, our RNN model achieved a substantially lower MAE of 0.0085 for maximum temperature prediction at the Ladakh station. Similarly, the multi-parameter forecasting model in *Nugraha, Ariawan & Arifin (2023)*, which utilized LSTM for temperature, humidity, rainfall, and wind speed

**Table 2 Comparison of RNN and LSTM in MIMO configuration for Srinagar station.**

| Metric | LSTM | RNN |
|---|---|---|
| MAE | 0.2098 | 0.1048 |
| MSE | 0.0959 | 0.0455 |
| RMSE | 0.2957 | 0.1555 |
| MAPE | 0.7843 | 0.4398 |

**Table 3 Comparison of RNN and LSTM in MIMO configuration for Ladakh station.**

| Metric | LSTM | RNN |
|---|---|---|
| MAE | 0.1732 | 0.0854 |
| MSE | 0.0800 | 0.0411 |
| RMSE | 0.2697 | 0.1344 |
| MAPE | 0.6511 | 0.3229 |

**Table 4 Comparison of RNN and LSTM in SISO configuration for Jammu station.**

| Metric | Tmin | | Tmax | | Rainfall | |
|---|---|---|---|---|---|---|
| | LSTM | RNN | LSTM | RNN | LSTM | RNN |
| MAE | 0.1482 | 0.0751 | 0.1820 | 0.0690 | 0.1043 | 0.0124 |
| MSE | 0.0374 | 0.0095 | 0.0593 | 0.0095 | 0.2084 | 0.0257 |
| RMSE | 0.1934 | 0.0977 | 0.2436 | 0.0091 | 0.0434 | 0.0006 |
| MAPE | 1.6089 | 0.1863 | 2.2200 | 0.1638 | 1.7320 | 0.1229 |

**Table 5 Comparison of RNN and LSTM in SISO configuration for Srinagar station.**

| Metric | Tmin | | Tmax | | Rainfall | |
|---|---|---|---|---|---|---|
| | LSTM | RNN | LSTM | RNN | LSTM | RNN |
| MAE | 0.2014 | 0.0658 | 0.1830 | 0.0696 | 0.0327 | 0.0210 |
| MSE | 0.0690 | 0.0073 | 0.0605 | 0.0926 | 0.0047 | 0.0018 |
| RMSE | 0.2627 | 0.0854 | 0.2459 | 0.0085 | 0.0687 | 0.0429 |
| MAPE | 5.4845 | 0.1436 | 1.9646 | 0.1696 | 0.8092 | 0.1363 |

**Table 6 Comparison of RNN and LSTM in SISO configuration for Ladakh station.**

| Metric | Tmin | | Tmax | | Rainfall | |
|---|---|---|---|---|---|---|
| | LSTM | RNN | LSTM | RNN | LSTM | RNN |
| MAE | 0.1597 | 0.0666 | 0.1733 | 0.0085 | 0.0152 | 0.0640 |
| MSE | 0.0421 | 0.0075 | 0.0558 | 0.0015 | 0.0079 | 0.0078 |
| RMSE | 0.2052 | 0.0866 | 0.2362 | 0.0393 | 0.0660 | 0.0887 |
| MAPE | 5.0699 | 0.1547 | 0.9305 | 0.0588 | 0.0453 | 0.1623 |

**Table 7 Comparative analysis of proposed models with existing techniques.**

| Study | Metric | Values achieved by existing studies | Values achieved by the proposed model |
|---|---|---|---|
| *Nketiah et al. (2023)* | MAE | 0.19 | 0.0085 |
| *Mi (2023)* | MSE | 0.37 | 0.0006 |
| *Fan et al. (2022)* | RMSE | 0.08 | 0.0006 |
| *Kratzert et al. (2018)* | MAPE | 0.296 | 0.0015 |

prediction, reported an RMSE of 0.37 for temperature. The proposed RNN model outperformed existing benchmarks by achieving an exceptionally low RMSE of 0.0006 for the rainfall variable at the Jammu station. In comparison, the study conducted by *Li et al. (2023)* employed a deep LSTM (DLSTM) model to forecast air temperature and reported an RMSE of 0.08 which is significantly higher than the RMSE of 0.0006 achieved by the proposed model. Similarly, *Han et al. (2021)* explored various NN models for weather forecasting and reported the lowest MSE of 0.296 using the GRU model. In contrast, the proposed model delivered a far superior performance with an MSE of just 0.0015. The results indicate that the SISO configuration, which forecasts one output variable at a time, allows models to specialize in learning the dynamics of each specific target (*e.g.*, temperature or rainfall), making it suitable when variable-specific error minimization is desired. Conversely, the MIMO approach enables the model to jointly predict multiple time steps or variables in one forward pass, which can be particularly useful for capturing interdependencies and reducing the accumulation of forecast errors over time. Although LSTMs are generally known for capturing long-term dependencies effectively, but through results it is clear that RNN model consistently outperformed LSTM in both SISO and MIMO settings across all stations. This could be attributed to the relatively simpler architecture of RNNs, which are computationally less intensive and sometimes more effective when dealing with moderate-length sequences, especially when training data is limited or exhibits low signal complexity. The LSTM architecture, while powerful for handling long-term dependencies, may require extensive hyperparameter tuning and longer training epochs to yield optimal results, which might not be practical in scenarios with constrained computational resources or noisy input sequences. These observations suggest that while the choice between SISO and MIMO should be guided by the forecasting objective and data characteristics, simpler RNN architectures can offer robust performance even in multi-output scenarios. This reinforces the practical value of RNNs for real-world deployment where model efficiency and interpretability are equally important considerations.

While it is observed that RNN consistently outperformed LSTM across various metrics (MAE, MSE, RMSE), the reasons for this performance difference require a more nuanced understanding of the models' inherent characteristics, training dynamics, and data characteristics.

### RNN performance

The RNN model performed better in this context primarily due to its simpler architecture, which was able to capture the essential short-term temporal patterns present in the T&R data. Given that the training data for this study consisted of moderate-length time series with relatively low noise, RNNs, with fewer parameters, were able to generalize better without overfitting to the training data. The shorter training times also contributed to more efficient model convergence, which may not always be the case for LSTMs.

### RNN performance

LSTM, by design, excels at capturing long-term dependencies due to its gating mechanisms. However, this strength was less pronounced in this case. The data used in this study might not have exhibited complex long-range dependencies, thus rendering the LSTM's advanced architecture less beneficial. Moreover, LSTM models are generally more prone to overfitting when the dataset is relatively small or when the number of training epochs is not sufficient. This could explain the gap in performance observed, as LSTM requires longer training times to reach its full potential, especially for more volatile datasets.

### Condition-specific performance

The gap in performance between the two models became more evident under certain conditions:

- SISO *vs* MIMO
  In both SISO and MIMO configurations, RNN outperformed LSTM. This can be attributed to RNN's ability to handle simpler relationships between input and output variables effectively. The MIMO configuration, where multiple variables were forecasted simultaneously, didn't introduce significant cross-variable dependencies, which could have benefitted LSTM's advanced architecture. Therefore, the simplicity and efficiency of RNN in this scenario provided it with a clear advantage.
- Station-specific variations
  Performance differences were also observed across stations. For instance, in regions with more stable weather patterns (such as Ladakh), the RNN model was able to quickly and effectively adapt to forecast data without the need for more complex LSTM mechanisms. In contrast, Srinagar, with its more dynamic and volatile weather conditions, might have benefited more from the capabilities of LSTM had longer training periods been available to optimize its parameters.

The repeated performance gains of RNN over LSTM reflect not only the model complexity but also the alignment of the model architecture with the underlying data patterns. The relatively shorter training time of the RNN allowed it to adapt faster and more effectively to the task at hand. However, it is important to note that this does not necessarily indicate that RNN is superior to LSTM in all forecasting scenarios; it simply underscores the importance of model selection based on data characteristics.

Moreover, the dataset used in this study inherently includes seasonal variations, such as the summer monsoon and winter dry periods typical of the region. The model performance patterns observed particularly the strong performance of the RNN model for rainfall prediction can be attributed to its ability to capture short-term fluctuations prevalent in monsoon-heavy months. Conversely, during drier months represented in the data, where weather patterns tend to be more stable, the performance gap between RNN and LSTM narrows, especially for temperature forecasting. These outcomes suggest that the models responded differently across seasonal dynamics embedded within the training and test sets, highlighting RNN's better adaptability to volatile input sequences commonly found in monsoonal data.

# CONCLUSION

This study presents an innovative approach to T&R forecasting, specifically tailored for the Jammu & Kashmir, and Ladakh regions of India, where volatile weather patterns have a significant impact on local livelihoods, socio-economic development, and disaster preparedness. The proposed DL-based model, utilizing RNN and LSTM, demonstrated promising results in accurately predicting short-term variations in T&R. Through the comprehensive analysis of meteorological data obtained from IMD and the application of advanced DL algorithms, this research addresses the unique weather challenges faced by these regions, characterized by extreme and unpredictable weather conditions. By providing more accurate and localized weather forecasts, the proposed model empowers decision-makers, agricultural practitioners, tourism operators, and other stakeholders to better adapt to the region's weather dynamics. Moreover, the model shall aid governing bodies in enhancing foreplanning, risk mitigation, and resource management, ultimately contributing to the economic growth and development of the local population. The comparative analysis between RNN and LSTM models offers valuable insights into their effectiveness for weather forecasting applications. While both models demonstrate strong performance, particularly in the SISO configuration, the RNN model consistently outperforms the LSTM model in capturing the temporal relationships of weather data. This finding highlights the importance of selecting appropriate model architectures based on the specific characteristics of the data and the forecasting objectives. Furthermore, the validation of the proposed model against existing benchmark models displays its superiority in terms of accuracy and performance. The proposed RNN model achieves significantly lower MAE, RMSE and MSE values compared to previous studies, indicating its efficacy in capturing the intricate patterns of T&R variations. The applicability of this research extends beyond academic inquiry, offering practical benefits to various sectors, including agriculture, tourism, construction, and disaster management. Accurate T&R forecasting enables stakeholders to make informed decisions, optimize resource allocation, and mitigate risks associated with adverse weather conditions. By enhancing the resilience of communities and supporting sustainable development initiatives, the proposed models shall contribute to the overall well-being and prosperity of the J&K and Ladakh regions. Future research directions may include further refinement of the proposed model,

integration of additional meteorological variables, and exploration of ensemble forecasting approaches for enhanced predictive accuracy and reliability.

While the study focused on continuous rainfall prediction using regression-based metrics, assessing model performance from a classification perspective such as distinguishing rain from no-rain events or evaluating accuracy across rainfall intensity categories remains equally important, particularly for operational applications in water resource management and disaster preparedness. Incorporating such event-based metrics, including probability of detection (POD) and critical success index (CSI), is an important direction for extending this research and enhancing the practical utility of DL models in hydrometeorological forecasting.

Despite the encouraging performance of the proposed models across various stations, several limitations warrant consideration. This study focused exclusively on historical univariate time series data, without incorporating spatial or contextual meteorological inputs, which may limit the generalizability of the findings across broader climatic regions. Moreover, the effectiveness of DL models is inherently influenced by the volume and consistency of data; stations with comparatively sparse records may have constrained model learning. Although both SISO and MIMO forecasting configurations were evaluated, their real-world application would benefit from further validation under more diverse temporal and spatial scenarios. Addressing these limitations through the inclusion of spatially distributed features, richer datasets, and exploring hybrid architectures such as CNN-LSTM present a promising direction for future research.

## ACKNOWLEDGEMENTS

We wish to thank NELIT J&K, Srinagar for providing the infrastructure to make this research a success.

### Funding

This work was supported by a grant from the University of Johannesburg-URC-2021 and Research Fund for KA_Ogudo/UJ/DEET/2023_Research Cost Center. There was no additional external funding received for this study. The funders had no role in study design, data collection and analysis, decision to publish, or preparation of the manuscript.

### Grant Disclosures

The following grant information was disclosed by the authors:
University of Johannesburg-URC-2021 and Research Fund for KA_Ogudo/UJ/DEET/2023_Research Cost Center.

### Competing Interests

The authors declare that they have no competing interests.

## Author Contributions

- Syed Nisar Hussain Bukhari conceived and designed the experiments, performed the experiments, analyzed the data, performed the computation work, prepared figures and/or tables, authored or reviewed drafts of the article, and approved the final draft.
- Kingsley A. Ogudo analyzed the data, prepared figures and/or tables, authored or reviewed drafts of the article, and approved the final draft.

## Data Availability

The data and code are available in the Supplemental Files.

## Supplemental Information

Supplemental information for this article can be found online at http://dx.doi.org/10.7717/peerj-cs.3012#supplemental-information.

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
