# Peer review of "Forecasting temperature and rainfall using deep learning for the challenging climates of Northern India"

_PeerJ Computer Science, doi:10.7717/peerj-cs.3012_

## Round 0.1 · original submission · Major Revisions

Please pay particular attention to the PDF provided by Reviewer 1.

Reviewer 1 ·

Basic reporting

Please see attached file

Experimental design

Please see attached file

Validity of the findings

Please see attached file

Additional comments

Please see attached file

Annotated reviews are not available for download in order to protect the identity of reviewers who chose to remain anonymous.

Reviewer 2 ·

Basic reporting

Recommendation: Major Revision

Addressing these suggestions will significantly improve the manuscript's overall quality and increase its chances of meeting the necessary standards for publication.
Abstract

• Explain why RNN outperforms LSTM despite LSTM’s recognized strength in capturing long-term dependencies.

Introduction
• The introduction is too broad and focuses on geographic details; they are too much for an AI paper to reduce to emphasize climatic importance.
• Explain the "T & R" abbreviation earlier and use the terms consistently throughout the manuscript.
• Bridge paragraphs better and make transitions clear; improve flow from geography to forecasting.
• Eliminate redundancy in climatic descriptions; much of the altitude data could be summarized more concisely.
• Better contrast between the limitations of traditional forecasting methods and DL usage will strengthen the justification for DL usage. To further improve introduction, and method sections consider these studies; (1) Artificial Intelligence and Numerical Weather Prediction Models: A Technical Survey, (2) Potential of artificial intelligence-based techniques for rainfall forecasting in Thailand: a comprehensive review.(3)A critical review of RNN and LSTM variants in hydrological time series predictions.

Experimental design

• Rewrite preprocessing steps to separate them into distinct subsections with rationale statements.
• Reasoning or validation-based reasoning to justify the choice of window size (7) for sequence generation in literature.
• The methods section should include performance metrics used for model evaluation for reproducibility and transparency.

Validity of the findings

Explain why RNN performed better than LSTM even though LSTM has been proven to have better capabilities in dealing with long-term dependencies.

Also included are statistical significance tests to confirm performance differences between RNN and LSTM models.

Integrate visualizations summarizing model comparison across stations for more straightforward interpretation.

Provide an acronym definition of MIMO and SISO at first mention for clarity to interdisciplinary readers.

Please give reasons for hyperparameter choices and how they affect observed performance differences.

---

## Round 0.2 · accepted · Accept

Given that the authors have satisfactorily responded to the revision requests, I recommend the manuscript be accepted for publication.

Reviewer 2 ·

Basic reporting

Dear Editors,
I hope this message finds you well.

Authors made said changes as per the given directions. I am happy to recommend and accept this article in its current form.
Thanks

Experimental design

Accepted

Validity of the findings

Accepted

Additional comments

Accepted